

# pH regulates the formation of organosulfates and inorganic sulfate from organic peroxides reaction with dissolved SO₂ in aquatic media

Lin Du, Xiaofan Lv, Makroni Lily, Kun Li, Narcisse Tsona Tchinda

Environment Research Institute, Shandong University, Qingdao, 266237, China

*Correspondence to*: Lin Du (lindu@sdu.edu.cn) and Narcisse Tsona Tchinda (tsonatch@sdu.edu.cn)

**Abstract.** Organic peroxides (OPs) are an important component of dissolved organic matter (DOM), detected in various aquatic media. Despite their unique functions as redox agents in water ecosystems, the complete mechanisms and factors controlling their transformation are not explicitly established. Here, we evaluate the pH effect on the aqueous-phase reaction of three selected OPs (methyl hydroperoxide (MHP), peracetic acid (PAA) and benzoyl peroxide (BZP)) with dissolved SO₂.

Results show that due to the presence of hydroperoxyl group in their structures, MHP and PAA preferably form inorganic sulfate and organosulfate (methyl sulfate for MHP and acetyl sulfate for PAA) depending on the pH, while BZP exclusively forms organosulfate (benzoyl sulfate) in the pH range investigated. Moreover, it is seen that the ability for PAA to form inorganic sulfate relative to organosulfate is more pronounced, which is supported by a previous experimental observation. The effective rate constants of the transformation of these peroxides within pH 1 – 10 and 240 K – 340 K ranges exhibit

positive pH and temperature dependencies, and BZP is seen to degrade more effectively than MHP and PAA. In addition to the pH impact, it is highlighted that the formation of organic and/or inorganic sulfate strongly depends on the nature of the substituents on the peroxy function. Namely, PAA and BZP are more reactive than MHP, which may be attributed to the electron-withdrawing effects of -C(O)R (R = -CH₃ and -C₆H₅ for PAA and BZP, respectively) substituents that activate the peroxy function. The results further indicate that the aqueous-phase degradation of OPs can adequately drive the change in the

chemical composition of DOM, both in terms of organic and inorganic sulfate mass fractions.

## 1 Introduction

Organic peroxides (OPs) are an important contributor to the atmospheric oxidation capacity, as they act as reservoirs for alkoxy radicals and organic peroxy radicals (ROx) (Ehn et al., 2014; Allen et al., 2022). Besides their well-known implication in the oxidation of biogenic volatile organic compounds and anthropogenic precursors, they were shown to contribute to secondary

organic aerosols (SOAs) by forming an important link between sulfate and SOAs (Tkacik et al., 2012; Riva et al., 2015; Hettiyadura et al., 2017; Huang et al., 2020). Besides their role as important atmospheric oxidants, OPs are also known to regulate the distribution of HOx and ROx radicals through large scale vertical and horizontal transport (Mari et al., 2000; Ravetta et al., 2001). OPs were ubiquitously detected in atmospheric media including air and precipitation, fresh and sea water, surface waters and coastal environment (Sauer et al., 2001; Morgan and Jackson, 2002; Sun et al., 2021). Their sources are





varied, depending on their chemical compositions and properties (Liu et al., 2014). In aqueous media, OPs are produced by the reduction of ROx radicals and from fluorescent dissolved organic matter (DOM) by photogeneration, while other sources include partitioning from gas-phase to particle-phase (O'sullivan et al., 2005; Sun et al., 2021).

Primary sinks of OPs include $SO_2$ oxidation in cloud and rain droplets, uptake on water surfaces and water ice, mainly forming water-soluble organic compounds and secondary sulfates (Böge et al., 2006; Hua et al., 2008; Ignatov et al., 2011). Previous

studies have indeed shown that OPs are highly reactive towards dissolved $SO_2$ in submicron aerosol (O'sullivan et al., 1996; Liou and Storz, 2010), despite the incomplete detailed mechanisms and kinetics of their transformation. Such reactions in aqueous-phase are known to be driven by the change in solution pH. Specifically, bulk phase pH was shown to impact the rates of multiphase $SO_2$ oxidation towards aqueous-phase aerosol formation by altering the reactions mechanisms and modifying the particle morphology, viscosity and radiative effects (Turnock et al., 2019; Lei et al., 2022). While some of these

pH impacts have been explored in detail, namely in altering the aerosol radiative forcing (Turnock et al., 2019), the detailed effect on reaction mechanisms and kinetics remains not fully elucidated. The lack of comprehensive information on the pH impact on the mechanisms and kinetics of the OPs reactions towards $SO_2$ hinders the complete assessment of the importance of such interactions in aquatic environments, for example in the chemical composition of DOM.

Methyl hydroperoxide (MHP, $CH_3OOH$), peracetic acid (PAA, $CH_3C(O)OOH$) and benzoyl peroxide (BZP, $C_6H_5$-

$C(O)OO(O)C-C_6H_5$)) are typical examples of OPs both from natural and anthropogenic origin. MHP is the simplest and most abundant OP in the atmosphere, and is most prevalent in the remote marine environment (Heikes et al., 1996; O'sullivan et al., 1999). It is believed to form from the ozonolysis of alkenes and from biomass burning (Klippel et al., 2011; Zhang et al., 2012). PAA is one of the simplest OPs containing the carbonyl function. It is formed form the reaction between acetic acid and hydrogen peroxide in the presence of a strong acid catalyst. PAA is widely used as disinfectant, sterilizing agent, oxidizer,

polymerization catalyst and sanitizer in wastewater treatment and various industrial applications and is known as a powerful degrading agent for aqueous organic micropollutants (Luukkonen et al., 2015; Zhang and Huang, 2020; Ao et al., 2021; Kiejza et al., 2021). BZP is an antiseptic highly used for acne treatment and in chemical industry (Kircik, 2013; Brammann and Mueller-Goymann, 2019), and it can easily enter the human body via food intake and skin absorption, causing potential risks such as tissue damage and eventually acting as tumor promoter (Kozan et al., 2010; Ding et al., 2019). As a strong reactive

oxygen species, BZP can be reduced to benzoic acid, with the ability to inhibit the growth of some microorganisms in wheat flour and corn starch (Ding et al., 2019; Yu et al., 2022). Previous studies also mentioned the possibility of BZP thermolysis to yield benzoyloxy radicals and volatile benzene in the aqueous-phase (Tu et al., 1996; Wang et al., 2023).

Despite MHP, PAA and BZP are lowly soluble in water compared to hydrogen peroxide for example, they are relatively present in aquatic media (Meylan and Howard, 1991; O'sullivan et al., 1996; Sun et al., 2021). Some studies have investigated

the chemistry of these OPs through particle uptake, wet and dry deposition, thermal and photochemical decomposition, and in $SO_2$ oxidation (Tan et al., 2020; Xu et al., 2021; Ignatov et al., 2011; Allen et al., 2022), yet there are still deficiencies in understanding their full chemistry in aquatic media. The current study applies quantum chemical calculations techniques to



explore the aqueous-phase reactions of MHP, PAA and BZP with dissolved SO$_2$. The pH effect on the mechanism and kinetics of these reactions is evaluated and their implication in altering the chemical composition of DOM is assessed.

## 2 Methods

### 2.1 Quantum chemical calculations

Geometry optimizations and vibrational frequency analysis of all molecules and stationary points were performed using density functional theory based on the $\omega$B97XD/6-31++G(d,p) method under the harmonic oscillator-rigid rotor approximation (Chai and Head-Gordon, 2008; Elm, 2022). This method has been shown to adequately describe molecular clustering and reactions involving transition state (TS) configurations (Elm et al., 2017; Elm, 2019). The aqueous-phase was modelled using the continuum solvation model based on the solute electron density (SMD) at the $\omega$B97XD/6-31++G(d,p) level of theory. This model, based on an implicit treatment of solvent, is adequate for describing atmospheric processes (Ostovari et al., 2018; Xu and Coote, 2019), and was shown to outperform explicit water treatment especially in resolving for energy barriers (Chen et al., 2019). All $\omega$B97XD/6-31++G(d,p) calculations were performed using the Gaussian 09 package (Frisch et al., 2013), while single-point energy corrections on $\omega$B97XD/6-31++G(d,p) structures were calculated with the DLPNO-CCSD(T)/aug-cc-pVTZ method using Orca version 4.2.1 (Riplinger and Neese, 2013). Details on calculating the Gibbs free energy in aqueous-phase are given in the Supplement.

### 2.2 Kinetics

The kinetic analysis was performed based on the transition state theory (Truhlar et al., 1996). The reaction proceeds through collision between initial reactants, i.e., dissolved SO$_2$ (henceforth, simply denoted as S(IV) = SO$_2$•H$_2$O + HSO$_3^-$ + SO$_3^{2-}$) and the organic peroxide (OP), to form the reactant complex (RC) in equilibrium with the initial reactants, followed by a rearrangement through a TS configuration to form the product complex according to the following reaction:

$$\text{S(IV)} + \text{OP} \leftrightarrow \text{RC} \rightarrow \text{TS} \rightarrow \text{Product complex.} \qquad \text{(R1)}$$

The transition state theory approach to determine the bimolecular rate constant ($k$) of reaction (R1) under the pseudo-steady-state approximation considers two main terms and it is expressed as $k_{bim} = K_{eq}k_{reac}$, where $K_{eq}$ is the equilibrium constant of formation of RC and $k_{reac}$ is the unimolecular rate constant for the reaction of RC to the product complex, given respectively by the following equations:

$$K_{eq} = \frac{1}{c^0} \times \exp\left(-\frac{\Delta G_{eq}}{RT}\right), \qquad (1)$$

$$k_{reac} = \frac{k_B T}{h} \times \exp\left(-\frac{\Delta G^{\#}}{RT}\right), \qquad (2)$$

where $\Delta G_{eq}$ is the Gibbs free energy of RC formation, $c^0$ is the standard concentration, $h$ is Planck's constant, $k_B$ is Boltzmann's constant, $\Delta G^{\#}$ is the Gibbs free energy barrier separating the RC from the products, $R$ is the molar gas constant, and $T$ is the absolute temperature.





To account for low-barrier and barrierless processes in reaction (R1), namely the formation of the reactant complex, the contribution of molecular diffusion is considered based on the Collins-Kimball theory (Collins and Kimball, 1949) and the overall rate constant for reaction (R1) is then expressed as:

$$k_{overall} = \frac{k_{bim} \times k_D}{k_{bim} + k_D} ,$$ (3)

where $k_D$ is the steady-state Smoluchowski rate constant given as (Smoluchowski M, 1917):

$$kD = 4\pi R_{S(IV),PO} DNA ,$$ (4)

$R_{S(IV),OP}$ is the reaction distance between reactants S(IV) and OP, defined as the sum of their radii, $R_{S(IV)}$ and $R_{OP}$, respectively. $N_A$ is the Avogadro number and $D$ is the sum of reactants diffusion coefficients $D_{S(IV)}$ and $D_{OP}$ (Truhlar, 1985). The diffusion coefficient for a reactant is related to its radius in any medium of viscosity $\eta$ by the Stokes-Einstein approach (Einstein, 1905). For reactants S(IV) and OP in water, these diffusion coefficients are:

$$D_{S(IV)} = \frac{k_B T}{6\pi\eta R_{S(IV)}} \quad \text{and} \quad D_{OP} = \frac{k_B T}{6\pi\eta R_{OP}} .$$ (5)

The radii were calculated with the Multiwfn software by assuming spherical reactants (Lu and Chen, 2012).

## 3 Results and discussion

The reactions with OPs can be explored at three different pH ranges, according to the prevalent states of dissolved $SO_2$: $SO_2 \cdot H_2O$ at pH < 1.81, $HOSO_2^-$ at 1.81 < pH < 6.97 and $SO_3^{2-}$ at pH > 6.97. Details on the distribution of dissolved $SO_2$ are given in **Table S1**. From this distribution, the following reactions were investigated:

pH < 1.81            $SO_2 \cdot H_2O + OP \rightarrow$ Products , (R2)

1.81 < pH < 6.97     $HOSO_2^- + OP \rightarrow$ Products , (R3)

pH > 6.97           $SO_3^{2-} + OP \rightarrow$ Products . (R4)

### 3.1 MHP reaction with dissolved $SO_2$

At pH below 1.81, the reaction of dissolved $SO_2$ with MHP is driven by the interaction between $SO_2 \cdot H_2O$ and MHP to form $SO_2 \cdot H_2O \cdot MHP$. The Gibbs free energy surface of this reaction, along with the structures of all stationary states are given in **Fig. 1**. The formation of $SO_2 \cdot H_2O \cdot MHP$ is relatively endergonic at 298.15 K and standard concentration of 1 M. This complex rearranges to methyl sulfate clustered to water ($CH_3OSO_3H \cdot H_2O$), through formation of an S-O bond between the sulfur atom of $SO_2$ and the oxygen atom of MHP. Although this product complex is formed with substantial Gibbs free energy gain, ca. -54.73 kcal mol$^{-1}$ below the reactants, the process is separated by a high energy barrier located at 45.72 kcal mol$^{-1}$ above $SO_2 \cdot H_2O \cdot MHP$. This particularly high energy barrier is associated with the difficulty of the $SO_2 \cdot H_2O \cdot MHP$ system to form two new S-O bonds prior to $CH_3OSO_3H \cdot H_2O$ formation. The high energy barrier in this process is conducive to an extremely slow reaction, which occurs with a low overall rate constant of $1.81 \times 10^{-21}$ M$^{-1}$ s$^{-1}$ at 298.18 K. Hence, the MHP+$SO_2 \cdot H_2O$ reaction forming sulfate is likely without significant relevance in natural waters. This was expected since the uptake coefficient



of $SO_2$ into the particle-phase has been shown to be weakened at pH below 2 and other processes such as acid-catalyzed processes are expected to drive organosulfate formation instead (Wang et al., 2019).

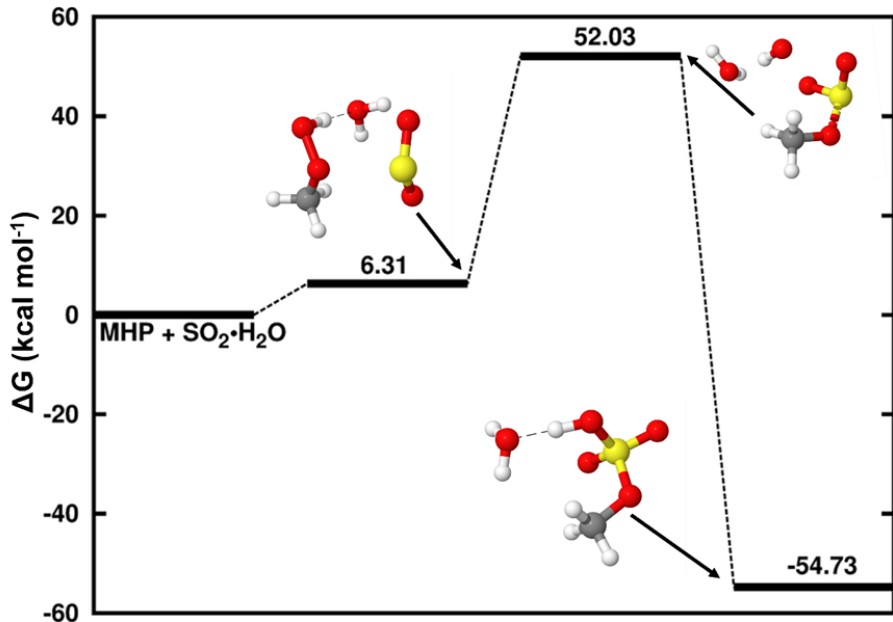

125

**Figure 1: Gibbs free energy profile of the stationary points in the MHP+SO₂•H₂O reaction. Atoms color coding is yellow for sulfur red for oxygen, grey for carbon and white for hydrogen.**

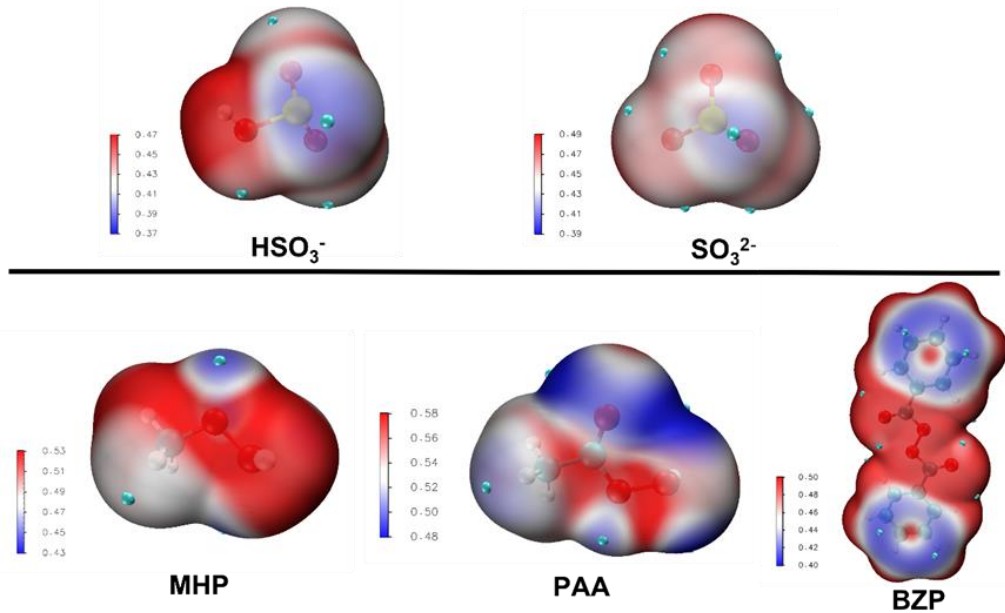

130 **Figure 2: Average local ionization energy (ALIE in a.u.) mapped molecular van der Waals surfaces of different reactants. The site possessing higher ALIE would attack the site which possesses lower ALIE, and vice-versa.**





At 1.81 < pH < 6.97 where the bisulfite ion is the prevalent state of dissolved $SO_2$, the mechanism of the MHP+$HOSO_2^-$ reaction is slightly different from that of MHP+$SO_2 \cdot H_2O$ reaction. From the average local ionization energy- (ALIE-) mapped molecular van der Waals surfaces of MHP and $HOSO_2^-$ shown in **Fig. 2**, the most probable reaction sites are oxygen atoms for MHP and the sulfur atom for $HOSO_2^-$. Hence, the sulfur atom of $HOSO_2^-$ can interact both with the oxygen atom of MHP connected to its carbon or that connected to its hydrogen atom, giving rise to two different configurations (RC1 and RC2, shown in **Fig. 3**) for MHP•$HOSO_2^-$. RC1 and RC2 react in different ways to form different types of sulfates according to the following reactions:

RC1 → TS1 → $CH_3OSO_3^-$ + $H_2O$ ,                  (R5)

RC2 → TS2 → $HSO_4^-$ + $CH_3OH$ .                   (R6)

In the two TS of reactions (R5) and (R6), the S atom of $HOSO_2^-$ is being connected to MHP through its $CH_3O$ and HO fragments, respectively, to form new S-O bonds. As a result, reaction (R5) forms organic sulfate (methyl sulfate, $CH_3OSO_3^-$) while reaction (R6) forms inorganic sulfate ($HSO_4^-$). The free energy barrier in reaction (R5) lies 36.02 kcal $mol^{-1}$ above RC1, being ~10 kcal $mol^{-1}$ lower than in the reaction at pH < 1.81 that also forms methyl sulfate. This is likely due to the electrostatic strain resulting from the electronic charge on $HOSO_2^-$, which induces a certain pronounced stability of the TS. This strain is also reflected in the stability of $CH_3OSO_3^- \cdot H_2O$, which is formed with -65.73 kcal $mol^{-1}$ free energy change.

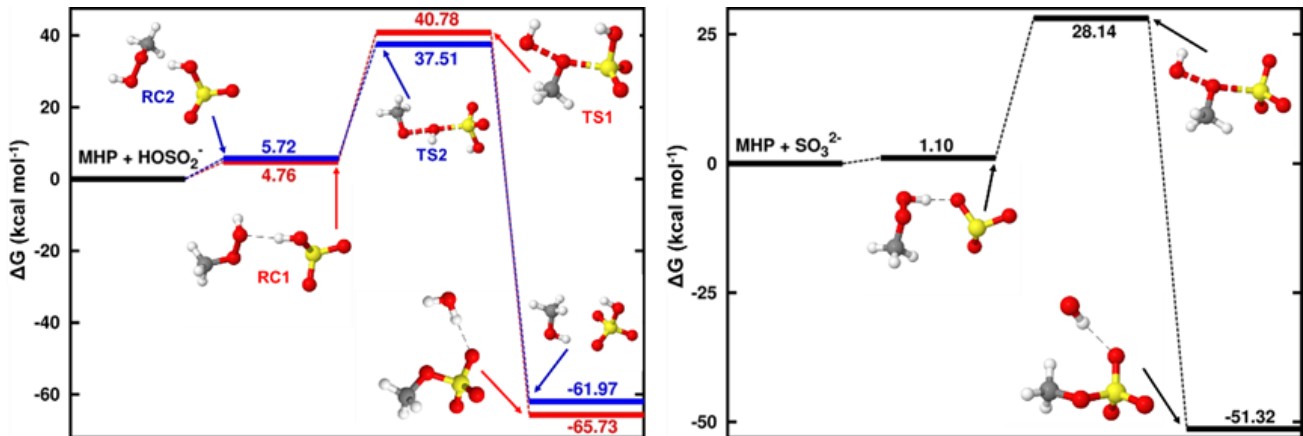

**Figure 3: Gibbs free energy profile of the stationary points in the MHP+$HOSO_2^-$ (left panel) and MHP+$SO_3^{2-}$ (right panel) reactions. Atoms color coding is yellow for sulfur red for oxygen, grey for carbon and white for hydrogen.**

Although RC1 and RC2 have almost the same formation Gibbs free energies (see **Fig. 3**), the free energy barrier heights in their respective paths are 36.02 kcal $mol^{-1}$ and 31.79 kcal $mol^{-1}$. This shows that at 1.81 < pH < 6.97, the process to form organosulfate is energetically less favored than the process to form inorganic sulfate. Our finding is in qualitative agreement with the results of Lind et al. who found, while investigating the aqueous-phase oxidation of S(IV) by MHP over the pH range 4.0-5.2, that 73% of inorganic sulfate were formed against 23% of methyl sulfate (Lind et al., 1987). The high proportion of



sulfate relative to methyl sulfate observed by Lind et al. can further be explained by the demonstrated fast hydrolysis of methyl sulfate at acidic pH (Hu et al., 2011) and its effective oxidation by OH radicals (Kwong et al., 2018) to form inorganic sulfate.

This further indicates that acidic waters may highly favor inorganic sulfate from MHP relative to organosulfate. Although the presence of additional water slightly alters the barrier towards sulfate formation (see **Fig. S1**), the overall effect of water is not significant due to poor water clustering to MHP. Further details are given in **Section S2**, in the Supplement.

As opposed to the bisulfite ion whose three oxygen atoms can feel different interactions from the sulfur atom, at pH > 6.97 when the ion is fully deprotonated, the three oxygen atoms of the sulfite ion ($SO_3^{2-}$) are identical and only one type of product,

namely organic sulfate is expected from its reaction with MHP (see **Fig. 3**). The reactant complex in this process is relatively more stabilized than RC1 and RC2, due to an increased electronic strain as a result of the presence of an additional electronic charge. The TS in this reaction lies 27.03 kcal mol$^{-1}$ above the reactant complex, being nearly 9 kcal mol$^{-1}$ lower than in the reaction that forms methyl sulfate in the pH range 1.81 - 6.97. This indicates that organosulfate formation is favored at high pH.

**3.2 PAA reaction with dissolved SO₂**

Due to the difficulty for SO₂•H₂O to form two new S-O bonds prior to organosulfate formation, the reaction with PAA was investigated at 1.81< pH < 6.97 and pH > 6.97, exclusively. As expected from the ALIE-mapped molecular van der Waals surfaces of (see **Fig. 2**), the reactive sites in PAA + S(IV) reactions are the oxygen atoms in the -O-O- function of PAA and the sulfur atoms of the S(IV) entities. Regardless of the pH range, both organic and inorganic sulfates are formed. Specifically,

the sulfur atom of dissolved SO₂ interacts with both oxygen atoms of the -O-O- function in PAA. Similar to the reaction with MHP, the sulfur atom attack on -O-O- through the oxygen atom connected carbon leads to the formation of an organosulfate (acetyl sulfate), while the attack through the oxygen atom connected to hydrogen gives rise to inorganic sulfate. Energetics and optimized structures of all relevant intermediates in PAA+S(IV) reactions are given in **Fig. 4**.

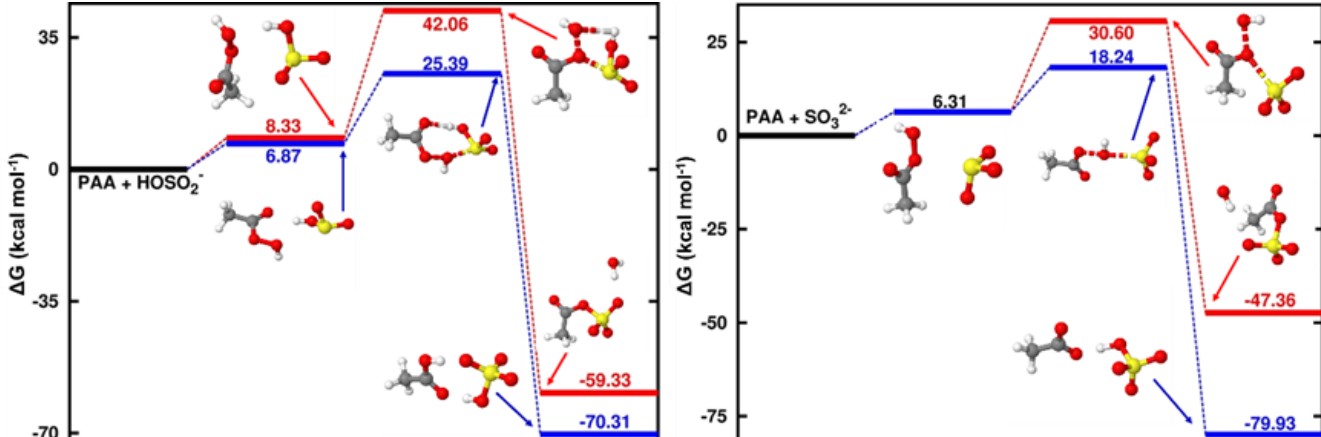

**Figure 4: Gibbs free energy profile of the stationary points in the PAA+HOSO₂⁻ reaction (left panel) and PAA+SO₃²⁻ reaction (right panel). Atoms color coding is yellow for sulfur red for oxygen, grey for carbon and white for hydrogen.**



In general, the reaction of PAA is much more favorable to the formation of inorganic sulfate than the reaction of MHP at all pH ranges, while the formation of organosulfate is slightly prevented. This is in line with the experimental observation that the

reaction of PAA with dissolved $SO_2$ almost exclusively forms inorganic sulfate (Lind et al., 1987). The transition states in the pathways of formation of inorganic sulfate are located at 25.39 and 18.24 kcal mol$^{-1}$ in the pH 1.81 – 6.97 and pH > 6.97 ranges, respectively, indicating a more favored process at basic pH than at acidic pH. The same trend is observed also in the pathways of organosulfate formation, with much higher energy barriers, in accordance with the reaction of MHP.

**3.3 BZP reaction with dissolved $SO_2$**

Unlike MHP and PAA that possess an hydroperoxyl group and, hence, capable of forming both inorganic and organic sulfate in its reaction with $HOSO_2^-$, BZP has two identical units from each side of the -O-O- group and that would exhibit similar reactivities in favor of organosulfate formation, exclusively. Based on the ALIE-mapped molecular van der Waals surface (**Fig. 2**), the oxygen atoms of the -O-O- function of BZP and the sulfur atom of $HOSO_2^-$ are the most likely reactive sites.


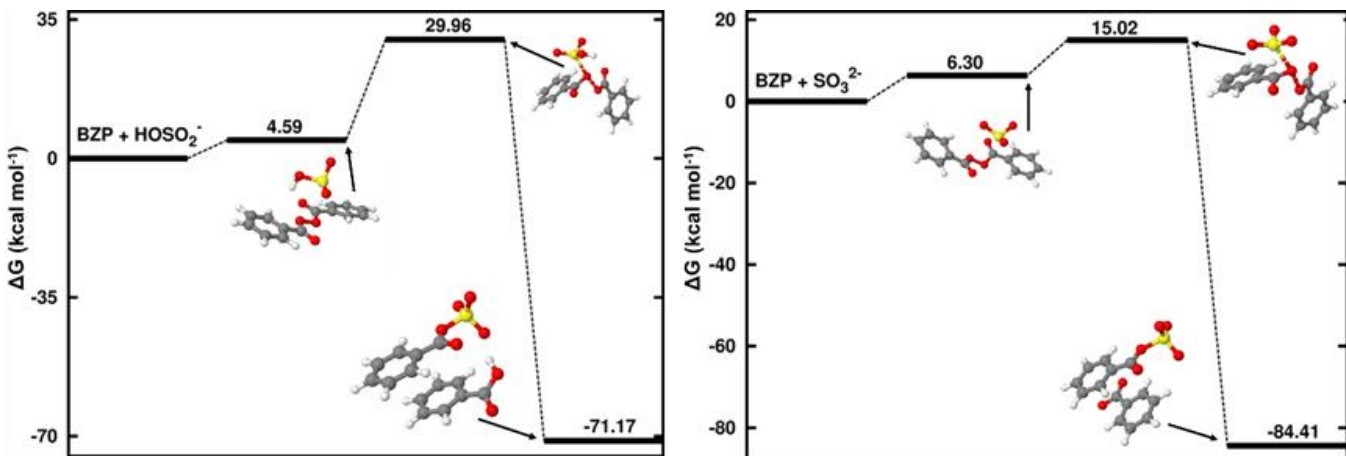

**Figure 5: Gibbs free energy profile of the stationary points in the BZP+$HOSO_2^-$ (left panel) and BZP+$SO_3^{2-}$ (right panel) reactions. Atoms color coding is yellow for sulfur red for oxygen, grey for carbon and white for hydrogen.**

The mechanisms of BZP reaction with S(IV) are nearly identical at all pH above 1.81, starting with the formation of a reactant complex that rearranges through a transition state configuration to form benzoyl sulfate (organosulfate), and benzoic acid. In the pH range 1.81 – 6.97, the TS is located at 25.37 kcal mol$^{-1}$ Gibbs free energy above the corresponding reactant complex (see **Fig. 5**). This is substantially decreased at pH > 6.97 with the transition state being located at 8.9 kcal mol$^{-1}$ Gibbs free energy above the reactant complex, indicating a more favored process at neutral to basic pH than at acidic pH. This favored

organosulfate formation at pH > 6.97 was also observed in the MHP+$SO_3^{2-}$ reaction, though less pronounced than in the BZP

 

reaction. Wang et al. detected both benzoyl sulfate and benzoic acid as the main products of the BZP+S(IV) reaction and proposed a mechanism in which the -O-O- function of the organic peroxide reacts directly with dissolved $SO_2$ (Wang et al., 2019). The current study confirms their experimentally proposed mechanism for organosulfate formation, with the further highlight that this mechanism might be more dominant at pH above 6.97.

**3.4 Reactions with the sulfite radical ion ($SO_3^-$)**

The importance of the sulfite radical ion ($SO_3^-$) to alter the atmospheric sulfur chemistry and to promote sulfate formation has been revealed in recent studies (Hung and Hoffmann, 2015; Hung et al., 2018). $SO_3^-$ is one of the main chain carriers of the atmospheric autoxidation of $SO_2$ commonly existing in atmospheric aquatic systems. The reactivity of $SO_3^-$ towards organic compounds to form organosulfur compounds has been investigated in a series of previous studies (Huang et al., 2020; Liu et

al., 2021). $SO_3^-$ can readily form from the reduction of $SO_3^{2-}$ by $NO_2$ according to the $NO_2(aq) + SO_3^{2-} \rightarrow NO_2^- + SO_3^-$ reaction catalyzed by both light and metal ions (Sapkota et al., 2015). Considering that $SO_3^-$ and $SO_3^{2-}$ share similar structural properties and may coexist in some aquatic environments, $SO_3^-$ reactions with MHP, PAA and BZP were also investigated. While our attempt to optimize PAA+S(IV) could not succeed, we found that $SO_3^-$ could outperform $SO_3^{2-}$ in the MHP reaction, decreasing the free energy barrier by 6.70 kcal mol$^{-1}$. However, the free energy barrier was substantially increased (by 18.31 kcal mol$^{-1}$)

in the BZP reaction instead. These reactivities differences of MHP and BZP towards $SO_3^-$ can be attributed to the repulsive effects of the electron density in the benzyl rings of BZP and the spin density of the free electron in $SO_3^-$, which seemingly destabilizes the transition state in the reaction with BZP.

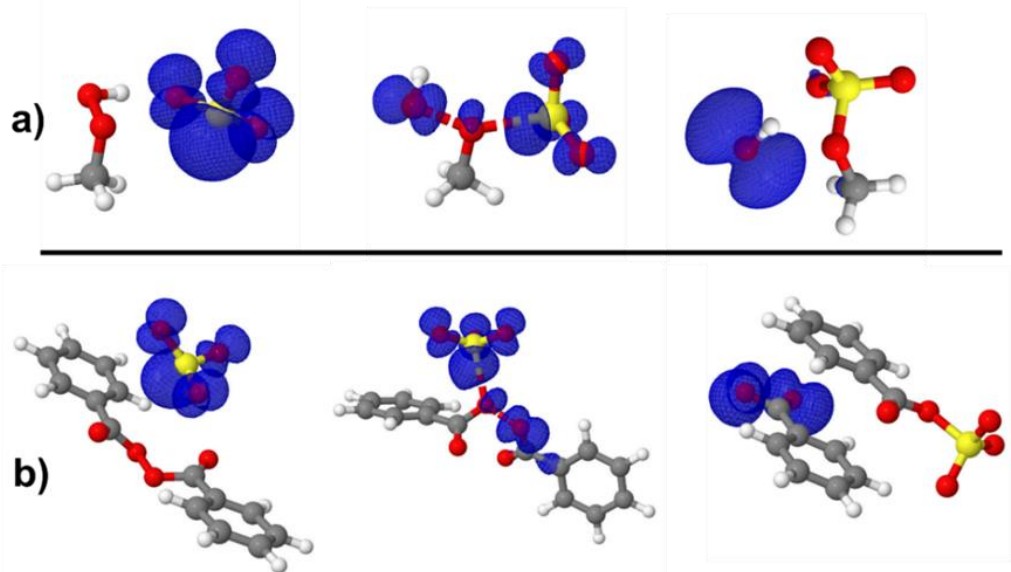

**Figure 6: Structures of all stationary states in the $SO_3^-$ reactions with (a) MHP and (b) BZP, including the**
**representation of spin density (in blue color). From left to right are reactant complex, transition state and product complex. The spin density indicates that the extra electron initially on $SO_3$, progressively migrates away from $SO_3$ as the reaction proceeds. Atoms color coding is yellow for sulfur red for oxygen, grey for carbon and white for hydrogen.**





A close inspection of the configurations of different stationary states in $SO_3^-$ reactions (**Fig. 6**) indicates that in the transition state of BZP reaction, the spin density is located opposite to the benzyl ring, which likely results in a strong repulsion. This

obviously destabilizes the transition state, giving rise to the high energy barrier as reported above. The situation, however, is different in the transition state of MHP reaction where the spin density is more relaxed in the configurational space of the main reactive entities.

## 4 Atmospheric implications

In a series of experiments, the Abbatt group suggested that in peroxides reaction with S(IV), organosulfates are formed directly

from peroxide–$SO_2$ interactions, instead of $SO_2$ being first oxidized by peroxides to $SO_4^{2-}$ followed by the reaction between $SO_4^{2-}$ and organic species (Ye et al., 2018; Wang et al., 2019; Wang et al., 2021). The current study is in line with the observations of Abbatt's group and further explains in details their suggested mechanisms for sulfate formation. Our calculations indicate that the importance of the degradation pathway of OPs by reaction with S(IV) to form (organo)sulfate in the aquatic environment is strongly dependent on the pH and is especially important under acidic pH. Wang et al. raised the

aerosol pH effect to be one of the fundamental issues to model aqueous-phase sulfate formation from the reactions of OPs with dissolved $SO_2$ (Wang et al., 2019). We explored the conditions of pH 1 – 10, that are dominated by different prevalent states of dissolved $SO_2$. While pH below 1.81 is not a favorable condition for the degradations of both investigated OPs by reaction with S(IV), these reactions exhibit different mechanisms and kinetics at pH above 1.81. Based on the mechanisms shown in **Figs. 1**, **3**, **4**, **5** and **S1**, the kinetics of the degradation of investigated OPs were evaluated within the ranges of pH 1 – 10 and

240 – 340 K temperature.

The reaction rate at each pH is driven by specific fractional populations of the different protonated states of S(IV): $\delta(SO_2 \cdot H_2O)$ at pH < 1.81, $\delta(HOSO_2^-)$ at 1.81 < pH < 6.97, and $\delta(SO_3^{2-})$ at pH > 6.97. Details on these fractional populations are given in **Table S1**. At a given temperature, the effective rate constant, $k_{eff}$, for each reaction is calculated by weighing the overall rate constant ($k_{overall}$, given by **Eq. (3)**) of partial reactions at specific $pH_i$ over the fractional populations of S(IV) ($x_{S(IV)}$) at that pH

as follows:

$$k_{eff,i} = \sum_i x_{S(IV),i} \times k_{overall,i} . \tag{6}$$

While $k_{overall}$ is independent of pH and simply gives the rate constant of OP + S(IV) reaction based on the activation energy, $k_{eff}$ gives the rate constant of the full process, by taking into account the population of different protonated states of S(IV) according to the pH value.

**Fig. 7** shows the effects of pH and temperature on the effective rate constants while the numerical values are given in **Table 1, Table S2 and Table S3**. In general, the effective rate constants of the reactions of OPs exhibit a positive temperature-dependency, with the steepest increase being observed at pH below 6.97, especially for the reaction of BZP. Above this range, the effective rate constant becomes almost insensitive to the pH. This indicates that the effect of pH change on the formation of organosulfates from reactions of OPs with dissolved S(IV) is most relevant in acidic aqueous environments.



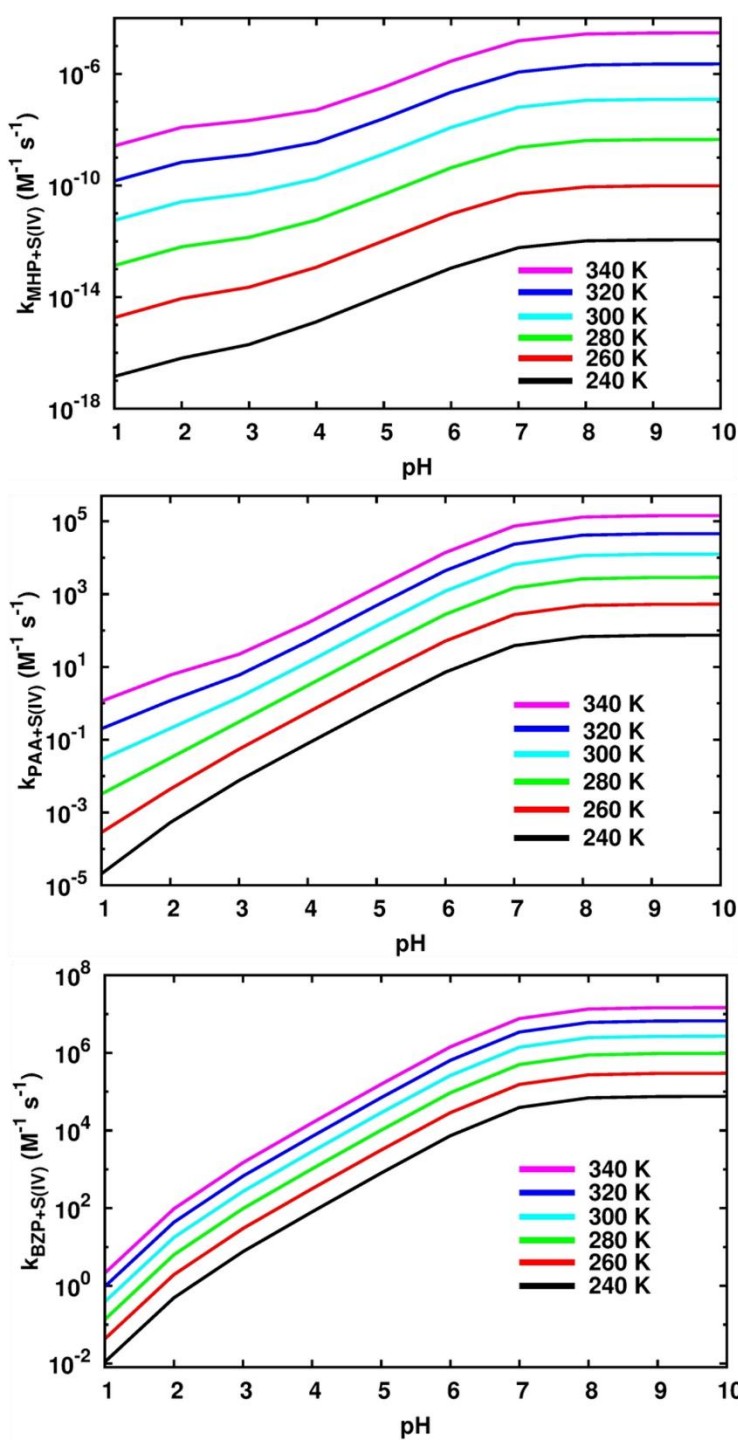


**Figure 7: Effective rate constants of S(IV) reaction with MHP (upper panel), PAA (middle panel) and BZP (lower panel) at different values of pH and temperature.**



**Table 1: Effective rate constants ($k_{eff}$, $M^{-1} s^{-1}$) for the reactions of MHP, PAA and BZP with S(IV) and atmospheric lifetimes (s) of corresponding OPs as a function of pH, at 298.15 K.**

| pH | MHP + S(IV) | | PAA +S(IV) | | BZP + S(IV) | |
|---|---|---|---|---|---|---|
| | $k_{eff}$ | lifetime | $k_{eff}$ | lifetime | $k_{eff}$ | lifetime |
| 1 | $4.07 \times 10^{-12}$ | $5.05 \times 10^{10}$ | $2.36 \times 10^{-2}$ | $8.71 \times 10^{0}$ | $3.54 \times 10^{-1}$ | $5.79 \times 10^{-1}$ |
| 2 | $1.90 \times 10^{-11}$ | $6.99 \times 10^{9}$ | $1.72 \times 10^{-1}$ | $7.73 \times 10^{-1}$ | $1.61 \times 10^{1}$ | $8.25 \times 10^{-3}$ |
| 3 | $3.77 \times 10^{-11}$ | $3.52 \times 10^{8}$ | $1.26 \times 10^{0}$ | $1.05 \times 10^{-2}$ | $2.48 \times 10^{2}$ | $5.34 \times 10^{-5}$ |
| 4 | $1.28 \times 10^{-10}$ | $1.03 \times 10^{7}$ | $1.19 \times 10^{1}$ | $1.11 \times 10^{-4}$ | $2.62 \times 10^{3}$ | $5.05 \times 10^{-7}$ |
| 5 | $1.01 \times 10^{-9}$ | $1.31 \times 10^{5}$ | $1.17 \times 10^{2}$ | $1.13 \times 10^{-6}$ | $2.61 \times 10^{4}$ | $5.07 \times 10^{-9}$ |
| 6 | $8.98 \times 10^{-9}$ | $1.48 \times 10^{3}$ | $1.07 \times 10^{3}$ | $1.24 \times 10^{-8}$ | $2.39 \times 10^{5}$ | $5.55 \times 10^{-11}$ |
| 7 | $4.79 \times 10^{-8}$ | $2.58 \times 10^{1}$ | $5.71 \times 10^{3}$ | $2.17 \times 10^{-10}$ | $1.28 \times 10^{6}$ | $9.69 \times 10^{-13}$ |
| 8 | $8.46 \times 10^{-8}$ | $1.46 \times 10^{-1}$ | $1.01 \times 10^{4}$ | $1.23 \times 10^{-12}$ | $2.26 \times 10^{6}$ | $5.48 \times 10^{-15}$ |
| 9 | $9.17 \times 10^{-8}$ | $1.35 \times 10^{-3}$ | $1.09 \times 10^{4}$ | $1.13 \times 10^{-14}$ | $2.44 \times 10^{6}$ | $5.06 \times 10^{-17}$ |
| 10 | $9.25 \times 10^{-8}$ | $1.34 \times 10^{-5}$ | $1.10 \times 10^{4}$ | $1.12 \times 10^{-16}$ | $2.46 \times 10^{6}$ | $5.02 \times 10^{-19}$ |


In the pH range 1.81–6.97, the rate constant of MHP + $HOSO_2^-$ reaction to form methyl sulfate (reaction (R5)) was estimated to be $2.38 \times 10^{-14}$ $M^{-1} s^{-1}$ at 298.15 K, ~three orders of magnitude lower than that of the reaction to form inorganic sulfate (reaction (R6); overall rate constant of $3.02 \times 10^{-11}$ $M^{-1} s^{-1}$), and 7 orders of magnitude higher than the overall rate constant of the $MHP+SO_2 \cdot H_2O$ reaction. As indicated above, this favored formation of inorganic sulfate relative to methyl sulfate is in

line with the experimental observation that inorganic sulfate is the major product while methyl sulfate is the second product in $MHP+HOSO_2^-$ reaction (Lind et al., 1987). Regardless of the temperature, the effective rate constant within this pH range increases by three orders of magnitude as the pH increases.

The formation of methyl sulfate at pH above 6.97 is much more favorable than in the preceding investigated pH range, occurring with an overall rate constant of $9.25 \times 10^{-8}$ $M^{-1} s^{-1}$ at 298.15 K. However, the effective rate constant in this range is

almost insensitive to pH at all temperatures investigated (see **Fig. 7**). The estimated atmospheric lifetimes for MHP at 298.15 K, based on this reaction, are $1.34 \times 10^{-5}$ s – $5.05 \times 10^{10}$ s in the pH range 1 – 10, and they are shown to decrease with increasing pH. This indicates that MHP degradation is more important in acidic waters and will alter the chemical composition of DOM by increasing the concentration of inorganic sulfate.

The kinetics of PAA+S(IV) reactions are essentially driven by the rates of inorganic sulfate formation, given the much lower

energy barrier than in organosulfate formation. The rate constants for these processes at 298.15 K are $9.37 \times 10^{-6}$ $M^{-1} s^{-1}$ and $1.10 \times 10^{4}$ $M^{-1} s^{-1}$ for organosulfate and inorganic sulfate formation, respectively. This is supported by the experimental observation from Lind et al. who found inorganic sulfate to be the sole product of the PAA+S(IV) reaction (Lind et al., 1987). They measured a rate constant of $(6.10 \pm 2.60) \times 10^{2}$ $M^{-1} s^{-1}$ for this reaction at 291 K and pH 2.9 – 5.8. At the same temperature and pH 5.8, we obtained an effective rate constant of $4.20 \times 10^{2}$ $M^{-1} s^{-1}$, in reasonable agreement with the experimental value.



At pH 1 – 10 and 240 K – 340 K, we obtained a positive dependency for the effective rate constant (plot and numerical values are shown in Fig. 7, Table 1 and Table S3). From on this kinetics, the estimated atmospheric lifetimes for PAA based on the reaction with dissolved $SO_2$ at 298.18K and pH 1 – 10 are within the range $1.12\times10^{-16}$ s – 8.71 s. This indicates that PAA is much more reactive than MHP towards dissolved $SO_2$ under investigated conditions.

  BZP is more reactive than MHP and PAA towards S(IV) at all pH above 1.81. Despite the mechanisms being nearly similar
in the investigated pH ranges, the overall rate constants of formation of benzoyl sulfate at 298.15 K are $1.55\times10^{-6}$ $M^{-1}$ $s^{-1}$ and $2.47\times10^6$ $M^{-1}$ $s^{-1}$ from BZP + $HOSO_2^-$ and BZP + $SO_3^{2-}$, respectively. The full pH and temperature dependencies of the total rate constant of BZP+S(IV) reaction are given in **Table 1** and plotted in **Fig. 7**. It should be noted that there is no direct formation of inorganic sulfate in BZP reactions in the pH range investigated. Sufficient evidence has been raised about aromatic organosulfates formation connected to anthropogenic activities, namely benzyl sulfate and phenyl sulfate detected in ambient
aerosols from urban sites (Kuang et al., 2016; Huang et al., 2018). These aromatic and other organosulfates derived from polycyclic aromatic hydrocarbons are believed to form from aromatics interacting with sulfur-containing species (Blair et al., 2017; Riva et al., 2015). It is likely that the investigated BZP+S(IV) mechanism in this work would explain some of such aromatic organosulfates formation mechanisms.

  The calculated lifetimes for benzoyl peroxide in the investigated pH range and 298.15 K are in the range $5.02\times10^{-19}$ – $8.25\times10^{-}$
$^3$ s (see **Table 1**), decreasing with increasing pH. These lifetimes are much lower than measured lifetimes of ~6 days in Los Angeles from photolysis of OPs (Krapf et al., 2016), indicating more efficient degradation in aqueous-phase. The current results highlight the strong pH impact on the degradation of BZP by S(IV). A previous study also showed that aerosol pH can substantially alter the formation of organosulfate from the reaction uptake of isoprene epoxydiols onto a mixture of ammonium sulfate and sulfuric acid particles (Lei et al., 2022).

The overall rate constants of $SO_3^-$ reactions with MHP and BZP were determined to be $7.66\times10^{-3}$ $M^{-1}$ $s^{-1}$ and $9.13\times10^{-8}$ $M^{-1}$ $s^{-1}$, respectively, regardless of the pH. Compared to the rate constants of S(IV) reactions, the pathway for $SO_3^-$ may also significantly contribute to degrade MHP and BZP, and form organosulfate. In general, sulfate formation from PAA+S(IV) and BZP+S(IV) reactions is more effective than from MHP+S(IV), and this pronounced reactivity can be attributed to electron-withdrawing effects of -C(O)R substituents (-C(O)-$CH_3$ and -C(O)-$C_6H_5$ in PAA and BZP, respectively) that activate the -O-
O- function. This finding further highlights that the specific nature of different substituents to the -O-O- function may play a determinant role in (organo)sulfate formation. It is obvious that OPs containing the hydroperoxyl function will contribute to both organic and inorganic sulfate mass, whereas those not containing this function will only form organosulfates. Although OPs may not compete with e.g., OH radicals in S(IV) oxidation, their contribution to sulfate formation may be important in oxidant-limiting conditions. The proposed mechanisms not only improve our understanding of the pH impact on the
degradation efficiency but also show that different OPs would alter the chemical composition of DOM in different ways, both in terms of organic and inorganic sulfate mass fractions. Of particular importance, the investigated pathways for the degradation of OPs have direct implication in aqueous-phase sulfate aerosol formation. This provides ground for deeper evaluation of the pH impact of the kinetics of organic and inorganic sulfate formation from various OP+S(IV) reactions for a



more complete kinetic data set. Such investigations will further guarantee a better understanding of the impacts of various

sources of sulfates in aqueous-phase aerosol formation and unveil a broader impact of OPs in altering the composition of DOM

in natural waters.

## Data availability.

All data from this research can be obtained upon request by contacting the corresponding author.

## Author contributions.

All authors contributed to the analysis and discussion of the results. NTT and LD designed the work. NTT performed all

quantum chemical calculations. LD and NTT wrote the paper with contributions from all co-authors.

## Competing interests.

The authors declare that they have no conflict of interest.

## Acknowledgements.

The authors would like to acknowledge the Wuxi Hengding Supercomputing Center Co., LTD for providing the computational

resources.

## Financial support.

This study was funded by National Natural Science Foundation of China (22076099) and Youth Innovation Program of

Universities in Shandong Province (2019KJD007).

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
