# Peer review of "pH regulates the formation of organosulfates and inorganic sulfate from organic peroxides reaction with dissolved SO2 in aquatic media"

_EGUsphere, 2023_

## Author Comment (AC1)

**Reply to Anonymous Referee #1**

We thank the Referee for their insightful comments on our manuscript. Here, we provide point-to-point response to all the comments. For clarity, the Referee's comments are reproduced in blue color text, authors' reply are in black color and modifications to the manuscript are in red color text.

In this work the authors carried out quantum calculations to explore the aqueous-phase reactions of organic peroxides (MHP, PAA and BZP) with dissolved SO2 or S(IV) species under various pH conditions. The simulated results clearly demonstrated the effects of pH on the major reaction pathways between various organic peroxides with S(IV) species leading to form inorganic sulfate and organosulfate. The potential reaction mechanisms were discussed with details and in general agreed with the existing experimental results. The paper is well written and provide greater mechanistic understanding of the organic peroxides chemistry in aspect of inorganic and organic sulfur formation.  I only have a few minor comments

General comments:

For the quantum chemical calculations, what are the concentrations of the reactants used in the calculations? Would the reaction pathways and kinetics  potentially affect by the reactant concentrations?

By its principle, quantum chemical (QC) calculations allow to solve for the electronic configuration of the reacting system and provide the reaction energies at all reaction states including the transition states, regardless of the reactants concentrations, and to analyze the reaction mechanisms of the developed reactions. This approach specially focuses on the electronic interaction between the reactants in order to provide atomic level insights into the processes driving the reaction. In this regard, QC calculations are mainly designed to predict and explain unknown reactions and are not affected by the reactants concentrations. Likewise, the transition states obtained from QC calculations are not dependent on the reactants concentrations.
By default, the rate constant of a reaction in aqueous phase (given by Eq. (3) in the main manuscript) is determined from QC calculations at a standard concentration of 1 M, with all other parameters being independent of the reactants concentrations. While the reactants concentrations do not affect the rate constants, they alter the overall kinetics by affecting the reaction rates, which are dependent on the concentrations. The rate constant is a fundamental parameter in kinetics that represents the constant of proportionality relating the reaction rate to the concentrations of reactants, while the reaction rate measures the change in reactant or product concentration per unit time.
The mention of 1 M standard concentration for computing the thermochemistry and the rate constant was already made in Section 2 of the main manuscript.

For the reaction pathways,  would there be any other possible reaction pathways between the organic peroxides and S(IV) in addition to the ones discussed and considered in the calculations?

From the average local ionization energy- (ALIE-) mapped molecular van der Waals surfaces of the reactants (shown in **Fig. 2** in the main manuscript), the most probable reaction sites are the -

O-O- functions of OPs and the S atom of dissolved $SO_2$. According to our QC calculations, the most likely outcome of OPs interacting with S(IV) is the formation of organic and/or inorganic sulfate, depending on the OP structure. Though we did not consider other reaction pathways else than abovementioned ones based on ALIE predictions, we speculate that any potential outcome of OP+S(IV) reaction would be without significance to the atmospheric environment. The following was added in the revised manuscript at page 6.

"In general, based on the ALIE-mapped molecular van der Waals surfaces, the OP+S(IV) reaction is mainly driven by the interaction between the -O-O- function of OP and the sulfur atom of S(IV), while other interactions are considered to be without significance."

In the atmospheric implications, it is very nice the authors to show and discuss the effective rate constants for different reaction systems under different values of pH and temperature and their corresponding lifetimes. the authors have pointed out such reactions may be important for sulfate formation under oxidant-limiting conditions. Can the authors further elaborate this point? Also, could the authors comment the yield of inorganic sulfate and organosulfates in different reaction systems and environmental conditions.

For decades, there has been considerable debate regarding the mechanisms responsible for atmospheric sulfate formation. Especially, under conditions of low solar radiation such as haze events where there is low production of OH radicals to oxidize $SO_2$, many studies have observed increasing sulfate formation and suggested alternative pathways for atmospheric sulfate (*Environ. Sci. Technol.*, 2022, **56**(15), 10608-10618; *Proc. Natl. Acad. Sci. U.S.A.*, 2020, **117**(3), 1354-1359; *Sci. Adv.*, 2016, **2**(12), e1601530). Besides sulfate formation mechanisms including ion-mediated and acid-catalyzed mechanisms that have already been elucidated, the current mechanisms can adequately account for sulfate formation under low OH radical conditions. This is further clarified in the revised manuscript. Moreover, to avoid misunderstanding "low OH radical conditions" is used instead of "oxidant-limiting conditions".

Page 13:
"These pathways can adequately account for sulfate formation under conditions of low solar radiation such as haze events where there is insufficient production of OH radicals."

Minor comments.

Line 100, "The diffusion coefficient for a reactant is related to its radius in any medium of viscosity $\eta$ by the Stokes-Einstein approach (Einstein, 1905)." It is not clear what is the viscosity of the aqueous solutions. How the diffusion would affect the reaction pathways if the solutions or aqueous aerosols were hlghly viscous?

The viscosity used here is the that of water. This is now specified in the revised manuscript:

Page 4:

"For reactants S(IV) and OP in water, the viscosity is that of water and the diffusion coefficients are calculated as:"

The overall rate constants of the studied reactions depend on the diffusion rate constants of the reactants in aqueous solution. According to its expression in Eq. (5) in the main manuscript, the diffusion rate constant is directly proportional to diffusion coefficients of the reactants, while the diffusion coefficient is inversely proportional to the viscosity of the solution (according to Eq. (5) in the revised manuscript). This shows that reactants in solutions with low viscosities will diffuse more easily than in solutions with high viscosities. Although the solution viscosity may not have an impact on the reaction mechanism, it definitely affects the reaction kinetics. We have highlighted the impact of viscosity in the revised manuscript.

Page 4:

"It is known that the viscosity of aerosols can be affected by relative humidity, chemical composition, water uptake and temperature (Song et al., 2021). Given that aerosols are mixtures of different proportions of organic and inorganic salts, it is obvious that the actual diffusion coefficients and overall kinetics of the studied reactions in aqueous aerosols will strongly depend on the actual aerosol viscosity."

Line 115, "The formation of SO2•H2O•MHP is relatively endergonic at 298.15 K and standard concentration of 1 M." Can the authors elaborate why the concentration of 1M was chosen for the calculations? Would the concentration of the reactants affect the reaction pathways and kinetics?

In QC calculations, standard conditions to calculate the Gibbs free energy in the gas-phase as defined in Gaussian are 298.15 K and 1 atm. For aqueous-phase reactions, standard conditions are defined in terms of absolute temperature and molar concentration, namely 298.15 K and 1 M. The expression of the Gibbs free energy in aqueous-phase, given in Eq. (S1) in the Supplement, includes a term that converts the standard pressure of 1 atm in the gas-phase to the standard concentration of 1 M in aqueous-phase. This was already given in the Supplement and is further highlighted in the revised manuscript.

Page 2:

"The Gibbs free energy in aqueous-phase is calculated at standard temperature of 298.15 K and molar concentration of 1M.  Details are given in Section S1 in the Supplement."

Moreover, as already explained in our reply to a previous comment of the Referee, while the concentrations of the reactants may not affect the chemical mechanism of the reaction, they will significantly affect the overall kinetics by altering the reaction rates.

Line 155, "The high proportion of sulfate relative to methyl sulfate observed by Lind et al. can further be explained by the demonstrated fast hydrolysis of methyl sulfate at acidic pH (Hu et al., 2011) and its effective oxidation by OH radicals (Kwong et al., 2018) to form inorganic sulfate." Could the authors comment how significance of these two processes in the formation of inorganic sulfate and organosulfates relative to the reactions between organic peroxides and S(IV) species?

From QC calculations, the two predicted pathways in the reaction of MHP with dissolved $SO_2$ are the formation of methyl sulfate and inorganic sulfate, with the path for inorganic sulfate being preponderant, in agreement with the experimental observation of Lind et al. (J. Geophys. Res.: Atmos., 92, 4171-4177, 1987). Considering that hydrolysis at acidic pH and OH-initiated reaction

of methyl sulfate are fast, the ultimate outcome of MHP + S(IV) reaction is nothing but inorganic sulfate formation. This implies that the reaction of OPs with dissolved $SO_2$ will mostly contribute to inorganic sulfate formation and the implication of this reaction in aqueous aerosol will be more pronounced in terms of inorganic sulfate mass loadings. This was already mentoned in Section 4 and is further highlighted at page 7.

Page 7:
"The implication for this is that MHP reaction with S(IV) will contribute to aqueous aerosol mostly in terms of inorganic sulfate mass loading."

Line 183, "In general, the reaction of PAA is much more favorable to the formation of inorganic sulfate than the reaction of MHP at all pH ranges, while the formation of organosulfate is slightly prevented. This is in line with the experimental observation that the reaction of PAA with dissolved SO2 almost exclusively forms inorganic sulfate (Lind et al., 1987)." What would be the ratios of the inorganic sulfate to organosulfate formed upon the reaction of PAA and dissolved SO2 at pH 1.81 – 6.97 and pH > 6.97? Also, what would be ratios for other reaction systems?

The kinetics of the reaction of PAA with S(IV) show that formation of inorganic sulfate is 11 orders of magnitude and 9 orders of magnitude faster than organosulfate formation at pH 1.81 – 6.97 and pH > 6.97, respectively. From the calculated rate constants, the branching ratios can be estimated to be 100% for inorganic sulfate formation and 0% for organosulfate. For the MHP reaction, the branching ratios are 99.92 % and 0.08% for inorganic sulfate and organosulfate, respectively. We have revised the above text and included the branching ratios in the revised manuscript.

Page 8:
"In general, the reaction of PAA is much more favorable to the formation of inorganic sulfate than the reaction of MHP at all pH ranges, while the formation of organosulfate is substantially prevented."

Page 13:
"The branching ratios estimated from the obtained the kinetics of MHP + S(IV) are 99.92% and 0.08% for inorganic sulfate and organic sulfate, respectively."

Page 13:
"Despite quantum chemical calculations predict the pathways for the formation of both inorganic sulfate and organosulfate from PAA+S(IV) reaction, the kinetics show that inorganic sulfate formation is more than 9 orders of magnitude faster than organosulfate formation in the whole pH range investigated, simply indicating that organosulfate formation from PAA reaction is insignificant. Regardless of the pH range, the branching ratios estimated from the kinetics are 100% for inorganic sulfate formation and 0% for organosulfate. This is in line with the experimental observation that the reaction of PAA with dissolved $SO_2$ almost exclusively forms inorganic sulfate (Lind et al., 1987)."

---

## Author Comment (AC2)

**Reply to Anonymous Referee #2**

We thank the Referee for their insightful comments on our manuscript. Here, we provide point-to-point response to all the comments. For clarity, the Referee's comments are reproduced in blue color text, authors' reply are in black color and modifications to the manuscript are in red color text.

This new contribution explores theoretically the aqueous condensed phase chemistry of organic peroxides (Ops) with dissolved sulfur in its (+IV) oxidation state as a function of pH. It especially simulates the chemistry of three selected OPs (methyl hydroperoxide (MHP), peracetic acid (PAA) and benzoyl peroxide (BZP)) with dissolved $SO_2$. This is certainly an important topic as organic peroxides and $SO_2$ are key components of aerosols and hydrometeors, that fits the scope of the journal.

Nevertheless, I would recommend that authors comments (and eventually modify their manuscript) to address the following comments, in addition to polishing the use of the English language.

We have addressed the Referee's comments and further polished the English in the whole manuscript.

While this reviewer is not an expert in the theoretical calculations reported here, the experimental section seems nevertheless short and not necessarily providing the level of information required to really assess the quality of the calculations.

We agree with the Referee that little is reported about experimental work. Originally, the paper is based on a theoretical study, exclusively, and we only referred to experimental results to support our findings. More has been added on the comparison with experiments in the revised manuscript.

Page 13:
"The branching ratios estimated from the obtained kinetics of MHP + S(IV) reaction are 99.92% and 0.08% for inorganic sulfate and organic sulfate, respectively. This favored formation of inorganic sulfate relative to methyl sulfate is in agreement with the experimental results that observed that inorganic sulfate is the major product while methyl sulfate is the second product in MHP+$HOSO_2^-$ reaction (Lind et al., 1987)."

Page 13:
"Despite quantum chemical calculations predict the pathways for the formation of both inorganic sulfate and organosulfate from PAA+S(IV) reaction, the kinetics show that inorganic sulfate formation is more than 9 orders of magnitude faster than organosulfate formation in the whole pH range investigated, simply indicating that organosulfate formation from PAA reaction is insignificant. Regardless of the pH range, the branching ratios estimated from the kinetics are 100% for inorganic sulfate formation and 0% for organosulfate. This is in line with the experimental study that found inorganic sulfate to be the sole product of the the reaction of PAA with dissolved $SO_2$ (Lind et al., 1987)."

The core content of this study concerns the effect of pH on the chemistry between the selected Ops and dissolved $SO_2$. This seems to be mainly (or even uniquely) simulated through the change of S(IV) species in presence at the selected pH. However, pH is known to catalyze the chemistry, and even the degradation, of Ops. This would certainly also affect the nature of reported transition states. But this not mention and corresponding papers not cited. Enami reported several studies, on different OPs, describing their acid catalyzed degradation (https://doi.org/10.1002/ejoc.202100343) or their overall fate in the condensed phase (https://doi.org/10.1021/acs.jpca.1c01513); while Krapf et al discussed their overall stability (https://doi.org/10.1016/j.chempr.2016.09.007).

The aim of this study was indeed the pH effect on the reaction of OPs with dissolved $SO_2$ towards sulfate formation, using quantum chemical (QC) calculations. By its principle, (QC) calculations allow to solve for the electronic configuration of the reacting system and provide the reaction energies at all reaction states including the transition states, regardless of the reactants concentrations, and to analyze the reaction mechanisms of the developed reactions. This approach specially focuses on the electronic interaction between the reactants in order to provide atomic level insights into the processes driving the reaction. In this regard, QC calculations are mainly concerned about how the electronic structures of the system evolve from the reactants to the products along the reaction process. While OPs do not dissociate with the change of pH, dissolved $SO_2$ adopts different protonated states depending on the pH and QC calculations allow to examine how each protonated state of dissolved $SO_2$ interacts with the OP to form the products. The average local ionization energy- (ALIE-) mapped molecular van der Waals surfaces of the reactants (shown in **Fig. 2** in the main manuscript) indicated that the most probable reaction sites for the reactants are the -O-O- functions of OPs and the S atom of dissolved $SO_2$. While hydronium ions have been shown to catalyze the degradation of hydroperoxides with α-hydroxyalkyl, α-alkoxyalkyl and α-acyloxyalkyl substituents (Environ. Sci. Technol. 2020, 54, 10561-10569; J. Phys. Chem. A 2020, 124, 10288-10295; J. Phys. Chem. A 2021, 125, 4513-4523), the available proton is unlikely to play a similar role in the degradation of OPs investigated in this study, due to unavailable α-substituent. In the series of Enami's studies on this topic, including theoretical calculations, the degradation of α-substituted hydroperoxides is guaranteed by the presence of the hydroperoxide group (-OOH) and the α-substituent with the oxygen atom (-O-) at the β-position so as to enable the formation of a ring structure and further facilitate the decomposition. These two requirements are not met for OPs considered in this work. Nevertheless, in our future investigations, we will minutely examine the potential role of available proton in the degradation of OPs that have no substituents at the α-position. The following was added in the revised manuscript to highlight our reasoning.

Pages 10-11:

"Besides reactions with dissolved $SO_2$, other recent studies showed that hydronium ions effectively catalyze the decomposition of organic peroxides at acidic pH (Qiu et al., 2020; Hu et al., 2020; Enami, 2021). It should be noted that in these studies, the catalytic effect of the available proton was examined on specific classes of OPs, namely hydroperoxides with α-hydroxyalkyl, α-alkoxyalkyl and α-acyloxyalkyl substituents. The presence of the hydroperoxide group (-OOH) and the α-substituent with the oxygen atom (-O-) at the β-position in these hydroperoxides enable the formation of a hydrogen-bonded ring structure involving the two functions and further facilitate their decomposition. Although these two requirements are not met for OPs considered in this work,

the potential role of the available proton in the degradation of OPs that have no substituents at the α-position is worth of thorough analysis in our future investigations."

In the opinion of this reviewer, it would be important to explore the effect of available protons on the actual structure of the transition state and not just on the distribution of S(IV) species.

As discussed in our reply to the Referee's previous comment, the configuration of investigated OPs is not in favor of explicit interaction of available proton. Hence, the transitions states were discussed solely in terms of interaction between the organic peroxide and the protonated state of S(IV) prevailing at the corresponding pH range.

The atmospheric implication is explored over a wide range of temperatures, corresponding even to ice conditions (at 240 K). It does not seem obvious that the temperature dependence of all parameters (such as acid-base equilibrium constants, etc.) have been considered to derive the temperature dependent rate constant. Could this be clarified? Also, this atmospheric significance needs to be compared to the lifetimes of the OPs which is also pH dependent (if the OP self-degrades faster that would limit the reported significance).

The temperature-dependent rate constant was derived by combining the transition state theory approach and the Collins-Kimball theory. The transition state theory is expressed by a temperature-dependent equation which is the product of two temperature-dependent terms: the equilibrium constant and the unimolecular rate constant for the reaction of the reactant complex to the product. Similarly, the Collins-Kimball theory is given by a temperature-dependent equation. These equations were already given in the manuscript and for further clarifications, the following modification was performed in the revised manuscript to highlight the equation for the transition state theory:

Page 3:
"The transition state theory approach to determine the bimolecular rate constant ($k$) of reaction (R1) under the pseudo-steady-state approximation considers two main terms and it is expressed as

$$k_{bim} = K_{eq} k_{reac} \,, \tag{1}$$

where $K_{eq}$ is the equilibrium constant of formation of RC and $k_{reac}$ is the unimolecular rate constant for the reaction of RC to the product complex, given respectively by the following equations:

$$K_{eq} = \frac{1}{c^0} \times \exp\left(-\frac{\Delta G_{eq}}{RT}\right) \,, \tag{2}$$

$$k_{reac} = \frac{k_B T}{h} \times \exp\left(-\frac{\Delta G^{\#}}{RT}\right) \,, \tag{3}$$"

Due to the simple structure of OPs investigated in this study relative to e.g., hydroperoxides with α-hydroxyalkyl, α-alkoxyalkyl and α-acyloxyalkyl substituents, proton-catalyzed self-degradation

is an unlikely process. Hence, the fate and estimated lifetimes of investigated OPs mainly depend on their reaction with S(IV). The following was added in the revised manuscript at page 11:

"Atmospheric lifetimes of OPs are calculated based on the effective rate constant of Eq. (7) and considering that self-degradation is an unlikely process given the simple structure of investigated OPs relative to those of e.g., hydroperoxides with α-hydroxyalkyl, α-alkoxyalkyl and α-acyloxyalkyl substituents whose self-degradations are catalyzed by the proton (Enami, 2021), these lifetimes are estimated based on S(IV) reactions, exclusively."

Minor points

The sentence starting line 30 (i.e., "In aqueous media, OPs are produced by the reduction of ROx radicals and from fluorescent dissolved organic matter (DOM) by photogeneration, while other sources include partitioning from gas-phase to particle-phase (O'sullivan et al., 2005; Sun et al., 2021).") does not provide a sound description of the formation of OPs.

This has been re-written as follows:

"Sources of OPs in aqueous media include self-reaction of photochemically formed organic radicals and partitioning from gas-phase to particle-phase (O'sullivan et al., 2005; Sun et al., 2021; Riemer et al., 2000)."

Line 33: " uptake on water surfaces", this is not a sink but rather a source.

To remove the confusion, the sentence has been re-written as:

"Primary sinks of OPs include $SO_2$ oxidation in cloud and rain droplets, mainly forming water-soluble organic compounds and secondary sulfates (Böge et al., 2006; Hua et al., 2008)."

Line 59, change lowly to poorly

This has been changed.

Line 122: the mention to uptake coefficients is unclear as bulk processes are described here.

We agree with the Referee this was not appropriate. The related sentence has been deleted in the revised manuscript.

**References**

Böge, O., Miao, Y., Plewka, A., and Herrmann, H.: Formation of secondary organic particle phase compounds from isoprene gas-phase oxidation products: An aerosol chamber and field study, Atmos. Env., 40, 2501-2509, 10.1016/j.atmosenv.2005.12.025, 2006.
Enami, S.: Fates of Organic Hydroperoxides in Atmospheric Condensed Phases, J. Phys. Chem. A, 125, 4513-4523, 10.1021/acs.jpca.1c01513, 2021.

Hu, M., Chen, K., Qiu, J., Lin, Y.-H., Tonokura, K., and Enami, S.: Temperature Dependence of Aqueous-Phase Decomposition of α-Hydroxyalkyl-Hydroperoxides, J. Phys. Chem. A, 124, 10288-10295, 10.1021/acs.jpca.0c09862, 2020.

Hua, W., Chen, Z. M., Jie, C. Y., Kondo, Y., Hofzumahaus, A., Takegawa, N., Chang, C. C., Lu, K. D., Miyazaki, Y., Kita, K., Wang, H. L., Zhang, Y. H., and Hu, M.: Atmospheric hydrogen peroxide and organic hydroperoxides during PRIDE-PRD'06, China: their concentration, formation mechanism and contribution to secondary aerosols, Atmos. Chem. Phys., 8, 6755-6773, 10.5194/acp-8-6755-2008, 2008.

Lind, J. A., Lazrus, A. L., and Kok, G. L.: Aqueous phase oxidation of sulfur(IV) by hydrogen peroxide, methylhydroperoxide, and peroxyacetic acid, J. Geophys. Res.: Atmos., 92, 4171-4177, 10.1029/JD092iD04p04171, 1987.

O'Sullivan, D. W., Neale, P. J., Coffin, R. B., Boyd, T. J., and Osburn, C. L.: Photochemical production of hydrogen peroxide and methylhydroperoxide in coastal waters, Marine Chem., 97, 14-33, 10.1016/j.marchem.2005.04.003, 2005.

Qiu, J., Tonokura, K., and Enami, S.: Proton-Catalyzed Decomposition of α-Hydroxyalkyl-Hydroperoxides in Water, Environ. Sci. Technol., 54, 10561-10569, 10.1021/acs.est.0c03438, 2020.

Riemer, D. D., Milne, P. J., Zika, R. G., and Pos, W. H.: Photoproduction of nonmethane hydrocarbons (NMHCs) in seawater, Marine Chem., 71, 177-198, 10.1016/S0304-4203(00)00048-7, 2000.

Sun, J., Ma, J., Lian, L., Yan, S., and Song, W.: Photochemical Formation of Methylhydroperoxide in Dissolved Organic Matter Solutions, Environ. Sci. Technol., 55, 1076-1087, 10.1021/acs.est.0c07717, 2021.